# *Caryocar coriaceum* Wittm. (Caryocaraceae): Botany, Ethnomedicinal Uses, Biological Activities, Phytochemistry, Extractivism and Conservation Needs

**DOI:** 10.3390/plants11131685

**Published:** 2022-06-25

**Authors:** José Weverton Almeida-Bezerra, José Jailson Lima Bezerra, Viviane Bezerra da Silva, Henrique Douglas Melo Coutinho, José Galberto Martins da Costa, Natália Cruz-Martins, Christophe Hano, Saulo Almeida de Menezes, Maria Flaviana Bezerra Morais-Braga, Antonio Fernando Morais de Oliveira

**Affiliations:** 1Department of Botany, Federal University of Pernambuco–UFPE, Recife 50670-901, Brazil; weverton.almeida@urca.br (J.W.A.-B.); josejailson.bezerra@hotmail.com (J.J.L.B.); viviane.silva@urca.br (V.B.d.S.); afmoliveira@gmail.com (A.F.M.d.O.); 2Department of Biological Chemistry, Regional University of Cariri–URCA, Crato 63105-000, Brazil; galberto.martins@gmail.com (J.G.M.d.C.); flavianamoraisb@yahoo.com.br (M.F.B.M.-B.); 3Faculty of Medicine, University of Porto, 4200-319 Porto, Portugal; 4Institute for Research and Innovation in Health (i3S), University of Porto, 4200-319 Porto, Portugal; 5Institute of Research and Advanced Training in Health Sciences and Technologies (CESPU), Rua Central de Gandra, 1317, 4585-116 Gandra PRD, Portugal; 6TOXRUN—Toxicology Research Unit, University Institute of Health Sciences, CESPU, CRL, 4585-116 Gandra, Portugal; 7Department of Biochemistry, Eure et Loir Campus, University of Orleans, 28000 Chartres, France; hano@univ-orleans.fr; 8Biotechnology Center, Federal University of Rio Grande do Sul–UFRGS, Porto Alegre 91501-970, Brazil; saulomenezes99@gmail.com

**Keywords:** oleic acid, Caryocaraceae, extractivism, flavonoids, Chapada do Araripe

## Abstract

*Caryocar coriaceum* is an endemic tree of Brazil, occurring mainly in the northeast region in the Cerrado environment. The species, popularly known as “pequi”, produces fruits that are used in the manufacture of oil for food and medicinal purposes. This work reviewed studies conducted with the species, highlighting its ethnomedicinal use, its pharmacological potential, including its chemical constituents, and its cultural and socioeconomic importance. Information was obtained through the main scientific research platforms. The keyword “*Caryocar coriaceum*” was used as the main index for searching the following platforms: PubMed^®^, PubMed Central^®^, SciElo, Scopus^®^ and Web of Science^TM^. The compiled papers demonstrate that *C. coriaceum* has great medicinal, economic and cultural importance for northeastern Brazil. Popularly, the fruits of *C. coriaceum* are used to treat broncho-pulmonary diseases (bronchitis, colds and flu). The fixed oil is widely used to relieve pain from various causes in the treatment of inflammation, flu, eczema, burns, fever, rickets, indigestion, heart murmurs, fatigue and erectile dysfunction. Some of these uses are corroborated by pharmacological trials, which have demonstrated the antioxidant, healing, anti-inflammatory, gastroprotective, antinociceptive and antimicrobial properties of the species. Chemically, fatty acids and phenolic compounds are the main constituents recorded for the species. Due to its medicinal properties, the fruits and oil of *C. coriaceum* have a high commercial demand and are one of the main forms of subsistence activities for local populations. On the other hand, the extractive practice of the fruits, associated with anthropic factors and its physiological nature, makes the species threatened with extinction. Thus, public management policies are highly necessary in order to avoid its extinction.

## 1. Introduction

Among the botany families occurring in Brazil, Caryocaraceae (1845) presents a total of 25 species distributed in Central and South America and comprises only two genera, *Anthodiscus* G. Mey. and *Caryocar* L. The species belonging to the genus *Caryocar* are found in varied phytogeographic domains, such as Amazonia, Cerrado or Savannah, Atlantic Forest and Caatinga or Seasonally Dry Tropical Forest [1,2].

*Caryocar coriaceum* Wittm. is a species that occurs mainly in the Caatinga. In this ecosystem, *C. coriaceum* was described and published in the Flora Brasiliensis in 1886 by the botanist Ludwig Wittmack (1839–1929) [3]. The first reports of the use of this species date from the 19th century [4,5]. The English botanist George Gardner (1810–1849), while passing through the Cariri region in the city of Crato (state of Ceará, northeast region of Brazil), reported that the fruit of “pequi” was used in popular cooking and pharmacopoeia. Its wood, of high quality, was used in the construction of mills [6,7].

The scientific literature shows that *C. coriaceum* is a species widely studied in academia, mainly because it is a well-known species. Thus, a review becomes necessary in order to demonstrate the state of the art for the species, as well as to compile information about the data described in the literature.

Since then, a culture about *C. coriaceum* has been established in the Cariri region, giving this species a high demand due to its versatility. This descriptive work aimed to compile the work done with the species, highlighting its ethnomedicinal uses, pharmacological potential, including phytochemistry, and its cultural and socioeconomic importance in northeastern Brazil.

## 2. Review

### 2.1. Botanical Aspects and Geographical Distribution

*C. coriaceum* is a species belonging to the family Caryocaraceae Voigt, nom. cons. and the order Malpighiales Juss. ex Bercht. & J.Presl [8]. The etymology of the genus is Greek, in which caryon means “core” or “nut”, while kara means “head”, referring to the globose fruit of the species. The specific epithet coriaceum refers to “leathery texture”, “thick” and “stiff”. The species is popularly known as “pequi”, “piqui” or “pequizeiro”, originating from the indigenous people of Pindorama (a territory of present-day Brazil) (py-qui), where py = skin or shell and qui = thorn, meaning “spiny shell”, arising from the thorns found in the epicarp of the fruit [9,10,11].

Morphologically, the individuals of *C. coriaceum* are arboreal, with stem lengths varying from 5 to 15 m in height. Its trunk reaches 35 cm in diameter and has a wood density of 0.78 g/cm^3^ (Figure 1). The trunks have thick bark and thick and angular branches, which can grow to the sides of the plant or close to the ground and which set the species apart from others in Cerrado areas [12,13,14].

The leaves of *C. coriaceum* are compound and trifoliolate, with opposite phyllotaxis. Each leaf has a petiole, which measures 1.5 to 4 cm long. The leaf margins can be serrate or crenate. The leaflets measure 3.7 to 7 cm long and 5 to 10 cm wide and are short petiolate, with an oval limb rounded to a slightly rectilinear apex. The base of the leaflets is subcuneate. The limb is glabrous, both on the abaxial and adaxial sides, with venation of the brochidodromous type and a coriaceous texture [15,16,17].

From the reproductive point of view, *C. coriaceum* presents inflorescences of the densiflorous raceme type that are 2.5 to 8.5 cm long. Each inflorescence presents from 10 to 16 flowers which are hermaphrodite. The flowers are composed of numerous stamens, reaching about 300 per flower. The calyx is composed of five reddish-green sepals and a dialipetal corolla containing five light-yellow petals. The gynecium has a tri or tetralocular globose ovary [18,19].

After pollination, which can be performed by self-pollination, birds (ornithophily) and bats (chiropterophily), a single individual of *C. coriaceum* is capable of producing 500 to 2000 fruits. These diaspores are of the ovoid drupe type, with dimensions varying from 4 to 7 cm in length and 6 to 8 cm in diameter and a mass varying from 100 to 220 g. The diaspores are formed by a light green coriaceous epicarp. The outer mesocarp is whitish, while the inner one is fleshy, with a color ranging from creamy yellow to intense yellow and sometimes orange. The endocarp, which protects the seed, is spiny and has a brownish color. Generally, each fruit has one to four lumps, also called putamen, depending on how many ovaries are fertilized and developed [15,17,19].

Phenological studies conducted in the Chapada do Araripe (Ceará state, Brazil) showed that the flowering of the species occurs from June to October, and the maturation of its fruits occurs from October to March [10,13,17,19].

Regarding its geographical distribution, C. coriaceum is native and endemic to Brazil, being found mainly in the northern region of the Brazilian Northeast in the states of Bahia, Piauí, Pernambuco, Maranhão and Ceará. In the latter, the presence of the species is more abundant due to its occurrence in areas of environmental protection (APA), such as the APA-Araripe and the Araripe National Forest (FLONA) (Figure 2). Among the municipalities of the Ceará state, there are collection records for Araripe (municipality number 4 on the map), Santana do Cariri (5), Nova Olinda (6), Crato (7), Barbalha (8), Missão Velha (9), Brejo Santo (11), Porteiras (12) and Jardim (13) [15,20,21,22,23,24,25,26,27].

In Chapada do Araripe, C. coriaceum occurs in Cerrado sensu stricto enclaves. This vegetation occurs at the top of the Chapada and is characterized by being a semideciduous savannah vegetation. It covers aluminized leached soils, which are responsible for the twisted branches and trunks of the plant species that occur in this environment, showing sinuses of hemicryptophytes, cryptophytes-geophytes, chamaephytes and phanerophytes, twisted with irregular branching, perennial or deciduous, sometimes with a well-developed cortex [28].

### 2.2. Ethnomedicinal Uses

Among the species with medicinal potential belonging to the genus *Caryocar* L., *C. brasiliense* Cambess., *C. villosum* (Aubl.) Pers. and *C. coriaceum* Wittm. stand out. These three species are popularly used for the treatment of swellings, respiratory diseases, wound injuries, gastric and inflammatory diseases, muscle pain and chronic arthritis [14,29].

Historically, the Kariri Indians of the Chapada do Araripe called individuals of *C. coriaceum* “*Pyrantecaira*” (i.e., which gives vigor and strength). Unfortunately, due to the colonization of the region by the Portuguese (1683–1713), the genocide of these peoples occurred, and along with them, some of the traditional knowledge associated with the species was lost [30,31].

Among the 67 plant species occurring in the Chapada do Araripe with medicinal potential, *C. coriaceum* is one of the most cited by rural communities, presenting 47 therapeutic indications. Ethnobotanically, almost all parts of *C. coriaceum*, except the flowers and roots, are used for the treatment of different types and diseases such as diseases of the skin and subcutaneous tissue, diseases of the eyes and annexes, diseases of the digestive system, diseases of the osteomuscular system and connective system, endocrine, nutritional and metabolic diseases, infectious and parasitic diseases, injuries, poisoning and some other consequences of external causes, diseases of the genitourinary system and diseases of the respiratory system [16,32].

The fruits of *C. coriaceum* are the most used organ in folk medicine, either in natura (pulp) or its derivatives, such as syrup or oil [33,34]. The internal mesocarp is consumed to combat broncho-pulmonary diseases (bronchitis, colds and flu) and tumors [35,36,37]. The fixed oil is widely used in the treatment of rheumatism, inflammation, muscle pain, sore throat, bronchitis, cough with secretions, flu, eczema, scalp disorders, lung pain, asthma, burns, fever, rickets, indigestion, heart murmur, wound healing, fatigue and erectile dysfunction [38,39,40,41,42].

Although medicinal indications focus on the fruits, other organs of *C. coriaceum* are also reported as therapeutic agents. The leaves, for example, are employed in the regulation of catamenial flow, while the bark is used to combat fever and as a diuretic [38,40]. According to Silva et al. [43], the leaves and fruits of *C. coriaceum* are used in the treatment of bronchitis, fatigue, nodule, catarrh, cicatrization, headache, toothache, sore throat, joint pain, mouth sore, influenza, broken bone, rheumatism and cough. In addition to the use for the treatment of human diseases, *C. coriaceum* leaves are employed in ethnoveterinary medicine to eliminate fetal attachments in cattle, and the fixed oil is applied to cuts and inflammations in various animals [31,44].

### 2.3. Biological and Pharmacological Activities

In addition to its ethnomedicinal uses, *C. coriaceum* has been scientifically evaluated for its medicinal potential. Due to its versatility and high rates of therapeutic indications, the fixed oils obtained from the fruits are the most investigated products by different authors [45,46,47,48,49,50,51]. Studies that have demonstrated the antimicrobial, healing, anti-inflammatory and gastroprotective antioxidant potential, in addition to other activities of *C. coriaceum*, are presented below.

#### 2.3.1. Antioxidant Activity

Among the various published studies on *C. coriaceum*, the antioxidant activity using the DPPH (2,2-diphenyl-1-picrylhydrazyl) free radical reduction method is the most investigated (Table 1). In this method, a total of 2.5 mL of *C. coriaceum* samples are added to a 1 mL solution of DPPH (60 µM), obtaining varying concentrations depending on the study, reaching a maximum of 1000 µg/mL. In addition, blank tests are used, which aim to measure the absorbance of the extraction solvents (ethanol, methanol or water). As a positive control, in this assay, known antioxidant agents are used, such as vitamin C (ascorbic acid) or BHT (butylhydroxytoluene). As a negative control, only the DPPH solution is used for comparative purposes. Finally, with the aid of a spectrophotometer, the absorbance of the solutions is measured after 30 min of reaction in an environment devoid of light. When the product shows antioxidant activity, the sample that has an original purple color tends to discolor to a yellowish color. So, the more yellow it is, the greater the antioxidant potential.

The aqueous extract of the leaves of the species, for example, showed the highest capacity in reducing DPPH, with an IC_50_ value 15 times lower than ascorbic acid, a positive control [52]. Duavy et al. [53] reported that, at concentrations of 100 and 250 μg/mL, the DPPH radical scavenging exhibited by *C. coriaceum* leaf and fruit peel extracts was similar to that found for ascorbic acid. Alves et al. [54] also demonstrated that the peels and pulp of *C. coriaceum* fruits showed antioxidant potential, with the pulp, with an IC_50_ value of 49.4 µg/mL, being the most active part.

In vivo antioxidative studies were conducted by Duavy et al. [50]. These authors found that the leaf extracts and oil from the pulp of *C. coriaceum* fruits conferred protection to *Drosophila melanogaster* (fruit fly) against the oxidant agent paraquat (1,1′-dimethyl-4,4′-bipyridinium dichloride). Methodologically, the researchers fed the flies with a sucrose solution (4%) containing paraquat (1 mM) and a group which was added to this solution: the aqueous extract of *C. coriaceum* leaves in varying concentrations (1–5 mg/mL), accompanied by a control (without paraquat and without extract). After a period of one week, the live flies were anesthetized on ice and manually homogenized in a phosphate buffer solution (20 mM), pH 7.4 (50 flies/mL), and the homogenate was centrifuged at 3500× *g* for 10 min at 4 °C. After centrifugation, the supernatant was collected and kept on ice until testing. Such procedures were carried out to determine the levels of reactive species in order to assess whether the product had an antioxidant effect. For this, the levels of reactive species were measured using the DCFDA assay, which is based on the deacetylation of 2′,7′-dichlorofluorescein diacetate (DCF-DA). The medium contained pH 7.4 potassium phosphate buffer (75 mM), DCFH-DA (5 µM final) and the fly supernatant (10 µL). Fluorescence was determined at 488 nm excitation and 525 nm emission, respectively, for 20 min in a Shimadzu spectrophotometer. The results were expressed as arbitrary fluorescence units (AUF), using a standard curve with DCF. In addition, lipid peroxidation was evaluated by measuring the levels of thiobarbituric acid reactive substances (TBARS) in the samples. In such an assay, the authors incubated the supernatant solution in an acidic medium (8.1% SDS (100 μL), 0.8% TBA (500 μL) and 20% acetic acid pH 3.5 (500 μL)) for an hour at 100 °C. TBARS were determined spectrophotometrically at 532 nm using malondialdehyde (MDA). The natural products were able to reduce the levels of reactive oxygen species (ROS) and lipidic peroxidation, as well as decrease the activity of the antioxidant enzymes catalase and glutathione-S-transferase. In addition, leaf and pulp extracts were able to down-regulate the mRNA expression of stress-related genes for catalase, superoxide dismutase, thioredoxin reductase and Keap-1 (Kelch-like ECH-associated protein 1).

**Table 1 plants-11-01685-t001:** In vitro antioxidant activity of *Cariocar coriaceum* using the free radical DPPH (2,2-difenil-1-picril-hidrazil).

Plant Organ	Preparation	IC_50_ µg/mL (Extract/Oil)	IC_50_ µg/mL (Positive Control)	Reference
Leaves	Aqueous extract	2.70	42.0 (Ascorbic acid)	[52]
Leaves	Ethanolic extract	3.24	42.0 (Ascorbic acid)	[52]
Leaves	Ethanolic extract	26.37	6.5 (Ascorbic acid)	[53]
Leaves	Aqueous extract	27.20	6.5 (Ascorbic acid)	[53]
Fruit peels	Ethanolic extract	38.66	6.5 (Ascorbic acid)	[53]
Leaves	Hydroethanolic extract	6.06	77.76 (Ascorbic acid)	[55]
Leaves	Methanolic extract	5.02	77.76 (Ascorbic acid)	[55]
Leaves	Hydroethanolic extract	9.70	6.20 (Butylated hydroxytoluene)	[56]
Fruit peels	Ethanolic extract	49.40	13.7 (Rutin)	[54]
Pulp	Ethanolic extract	25.50	13.7 (Rutin)	[54]
Pulp	Fixed oil	10.21	13 (Ascorbic acid)	[57]

#### 2.3.2. Antimicrobial Activity

The antimicrobial activity of products from *C. coriaceum*, especially its fixed oil, against bacteria of clinical interest has been determined by different authors. For the assays to determine the Minimum Inhibitory Concentration (MIC) of the antibacterial assays using the microdilution method, a serial dilution of the natural product (1024–1 µg/mL) in BHI broth (Brain Heart Infusion) was used, which contained 10% of the bacterial inoculum in suspensions of 10^5^ CFU/mL. Subsequently, the culture plates were placed in bacteriological incubators for incubation at 37 °C for 24 h. After this period, solutions (20 uL) of liquid resazurin were added to the plates in order to carry out the redox reactions. The MIC was defined in such studies as the lowest concentration at which there was no bacterial growth. Furthermore, in some studies, the researchers evaluated the ability of *C. coriaceum* to enhance the action of antibiotics against multidrug-resistant bacteria using the same methodology; however, the product concentrations were sub-inhibitory (MIC/8).

Saraiva et al. [48] demonstrated through microdilution assays that the oil presents a minimum inhibitory concentration (MIC) of 512 µg/mL against *Escherichia coli* ATCC 25922, *E. coli* EC 27, *Staphylococcus aureus* ATCC 12692 and *S. aureus* SA 358; however, it proved ineffective against *Pseudomonas aeruginosa* ATCC 15442 and *Proteus vulgaris* ATCC 13315. These authors also demonstrated that the fixed oil of *C. coriaceum* was able to reduce the MIC of aminoglycoside antimicrobials (gentamicin, kanamycin, amikacin and neomycin) against *E. coli* and *S. aureus*.

Costa et al. [47] found through the disc diffusion method that the fixed oil of *C. coriaceum* at a concentration of 10 µg/disc showed an antibacterial effect on *Salmonella choleraesuis* ATCC 13314 (15 mm halo), *S. aureus* ATCC 12692 (13.7 mm), *P. aeruginosa* ATCC 15442 (10.3 mm) and *Streptococcus pneumoniae* ATCC 6314 (7.7 mm). In the case of the agar diffusion method, the bacteria are replicated in Petri dishes containing MüllerHilton agar, using a sterile swab. After sowing, paper discs (6 mm in diameter) are impregnated with a solution (20 µL) of the natural product of *C. coriaceum* in different concentrations (1.25–10%), which are placed in the center of the plate agar. Ampicillin (AMP, 100 µg/disk) and chloramphenicol (CLO, 100 µg/disc) are used as positive controls, and Tween 80 and distilled water are used as negative controls. Subsequently, these plates were incubated at 37 °C for 24 h. Finally, the inhibition halos of each concentration were measured.

Despite the bactericidal effects of the fixed oil reported above, Pereira et al. [58] found no in vitro antibacterial action of the oil at concentrations of clinical relevance (1024 µg/mL) against the strains of *Proteus vulgaris* ATCC 13315, *Klebsiella pneumoniae* ATCC 10031, *Shigella flexneri* ATCC 12022, *P. aeruginosa* ATCC 9027, *E. coli* 06, *Bacillus cereus* ATCC 33018, *S. aureus* ATCC 6538 and *S. aureus* 10.

According to Araruna et al. [59], the leaf extracts of *C. coriaceum* have an antibiotic modifying action. In this study, the authors found that the hydroethanolic extracts and the methanolic fraction were able to enhance the activity of different aminoglycoside antibiotics against *E. coli* 27 and *S. aureus* 358. Lacerda-Neto et al. [56] also observed that the hydroalcoholic leaf extracts of *C. coriaceum* enhance the effect of penicillins, such as benzylpenicillin, against *E. coli*.

The antimicrobial effect of *C. coriaceum* also extends to fungal strains of veterinary interest, such as *Microsporum canis* and *Malassezia spp*. [54]. In such a study, spore suspension solutions were used for growth on potato dextrose agar (PDA) and placed in microbial growth ovens at 28 °C for a period of 7 days. After growth, the spores were quantified in a Neubauer chamber to reach a concentration of 10^5^ to 10^6^ cells. In 96-well microplates, a solution of 100 µL of RPMI medium was added, which was microdiluted with *C. coriaceum* extracts starting from a concentration of 2500 until reaching 2.44 µg/mL. Finally, 50.0 µL of the fungal suspension was added to all wells of the plates, except for the lines intended for the control of the sterile medium. The readings were made by verifying the MIC, with the aid of stereoscopic verification of the lowest concentration of the samples capable of inhibiting 100% of the growth of the microorganism, after 5 days of incubation. In this study, the authors demonstrated that the ethanolic extracts of the fruit peel and pulp are excellent antifungal agents, since both extracts showed an MIC of 4.88 µg/mL against *M. canis*.

In contrast, the antifungal effect was not observed against species of the *Candida* genus of clinical interest, such as *Candida albicans*, *Candida glabrata*, *Candida krusei* and *Candida tropicalis* [59,60].

#### 2.3.3. Healing and Anti-Inflammatory Activity

Quirino et al. [46] showed that the oil from the *C. coriaceum* pulp had a significant healing effect in rats on the sixteenth day after administration. Therefore, acute gastric lesions were induced in mice (*n* = 8/group) by the oral administration of absolute ethanol (96%) in a volume of 0.2 mL (using an orogastric metal tube), and *C. coriaceum* oil dissolved in Tween 80 (2% in distilled water as a vehicle) was administered in oral doses of 200 and 400 mg/kg, 60 min before ethanol application. The vehicle treated group (2% Tween 80) was included as a negative control. After a period of 30 min after the administration of the solvent, the animals were killed by cervical dislocation; following ethical procedures, the stomachs were removed and opened along the greater curvature, and the area of the gastric lesions was measured by planimetry using a transparent grid.

Oliveira et al. [61] also evaluated such effect on excisional skin lesions in mice. Methodologically, mice were treated with ointment containing the fixed oil of *C. coriaceum* at 6% (*v*/*w*) and 12% (*v*/*w*), while the control group consisted of mice treated with solid ointment (vaseline and lanolin—1: two). As a positive control, 5% *w*/*w* clostebol acetate and neomycin sulfate cream were used, and T5 received 0.9% saline as a negative control. After treatment, the mice were anesthetized by the open mask method, and their dorsal surface was scraped with a sterile blade. In this area, the holidays were left naked in the open environment, and daily observations were made. Treatments were applied topically once a day from wound induction to complete healing in sufficient amounts to cover all wounds. These authors showed that, on the seventh day of treatment, the 12% oil caused a contraction of up to 96.5% of the wounds compared to 88% of the control drug.

Batista et al. [40] added *C. coriaceum* oil to a base cream. To achieve the results, the authors divided mice into two groups. One consisted of 20 mice with skin wounds treated with the topical application of a cream base with 10% *C. coriaceum* oil. The second group included the same number of animals that received the topical application of the base cream without the natural product. After antisepsis and local anesthesia, a circular wound of 1 cm in diameter was surgically produced in the dorsum lumbar region. The skin lesions were evaluated under the clinical, morphometric and histological aspects on the 3rd, 7th, 14th and 21st postoperative days. At the end of the experiment, the skin wounds of the rats were fully healed, with complete closure of the edges, while the wounds of the animals in the control group still required more time for healing.

Regarding the anti-inflammatory potential, Oliveira et al. [61] showed that the fixed oil of *C. coriaceum* attenuated xylene-induced inflammatory edema in albino Swiss mice (*Mus musculus*) in a dose-dependent manner. The methods consisted of randomly allocating animals into nine groups. The first received only xylene (positive control), while the second received the fixed oil of *C. coriaceum* in natura. Another four groups received the oil in varying concentrations (6%, 12%, 25% and 50%), while the other two groups received dexamethasone (2.5 mg/kg) and indomethacin (5 mg/kg) orally for 3 days. One hour after the last oral treatment of mice, xylene was applied. Edema was induced by the topical application of xylene to the inner and outer surfaces of the right ear lobe. The left ear was considered as a control. Fifteen minutes or one hour after the induction of inflammation, the mice were sacrificed by an overdose of ether anesthesia, and both ears were removed. Circular sections were made using a cork drill with a diameter of 5 mm and weighed. The edematous response was measured as the difference in weight between the right and left ears. In this study, the crude oil reduced inflammation by 38.01% in just 15 min. According to these authors, the fixed oil of *C. coriaceum* accelerates the repair of cutaneous wounds, validating its popular use.

Similar results were also found by Saraiva et al. [62]. *C. coriaceum* oil at a concentration of 8 mg/ear was responsible for inhibiting 28.5% of *Croton* oil-induced inflammation in mice. After 48 h of application, a significant reduction in ear thickness compared to the group treated with the saline solution was also found.

Using the method of inflammation induced in the paw of mice by carrageenan, Figueiredo et al. [63] demonstrated that doses of 500 and 1000 mg/Kg of the fixed oil of *C. coriaceum* reduced the induced edema by 21% and 31%, respectively, after seven days of treatment. Methodologically, different groups of animals were treated with *C. coriaceum* oil at different doses (500, 1000 and 2000 mg/kg and 0.9% saline solution) for 7, 15 and 30 days, and at the end of each period, the initial volume (Vi) of the right paw was recorded using a plethysmograph. After 1 h, each animal received an intraplantar injection of 2% (*w*/*v*) carrageenan in the right hind paw (0.2 mL/paw). The volume of the right hind paw of each animal was evaluated again by the plethysmograph at 1, 2, 3 and 4 h after the injection of the phlogistic agent.

Silva et al. [15] verified, in rats, an anti-inflammatory action of the fixed oil after tendonitis induced with the intratendinous injection of collagenase in the calcaneal tendon. In this research, a total of 36 male rats were divided into groups: control, ultrasound associated with *C. coriaceum* oil and pure. In order to induce tendinitis, the intratendinous injection of collagenase was used in the right Achilles tendon. The treatment consisted of the daily application of ultrasound + oil or oil alone to the tendon. Macroscopic analysis was performed with a caliper on the 1st, 7th and 14th days. Subsequently, the rats were sacrificed, and then the tendon was dissected and removed to allow for histological analysis with Hematoxylin & Eosin (HE). Such action was evidenced in the reduction of neutrophils (inflammatory cells) after seven days of topical treatment with the oil. The researchers showed that when the tendinitis was submitted to ultrasonic waves, the process of tissue repair was more effective, because there was an induction of fibroblasts increase.

Oliveira et al. [49] induced knee arthritis in rodents using zymosan, a polysaccharide from the cell wall of *Saccharomyces cerevisiae* that produces acute and severe inflammation. In this research, ethyl acetate extract from the pulp of was used, which was diluted in 2% of Tween 80 concentrations and administered by oral gavage at doses of 100, 200 and 400 mg/kg 45 min before zymosan-induced arthritis or for 7 consecutive days at the same time each day. The zymosan injection was performed 24 h after the last administration of the extract. After the entire experimental protocol, the animals were then euthanized following ethical protocols for collecting fluid and synovial tissue to evaluate leukocyte recruitment, myeloperoxidase (MPO) activity and cytokine release and immunohistochemistry. Changes in vascular permeability were assessed by the extravasation of Evans Blue dye into joint tissue 6 h after zymosan injection. Thus, they demonstrated that the fixed oil of *C. coriaceum*, at doses of 100 mg/Kg, showed anti-inflammatory potential by reducing the influx of leukocytes and neutrophils into the joint cavity. In addition to the fruit, the leaves of *C. coriaceum* also show anti-inflammatory potential in mice. According to Araruna et al. [55], the hydroethanolic and methanolic leaf extracts of *C. coriaceum* reduced edema in Swiss mice (*Mus musculus*) caused by different sensitizing agents such as arachidonic acid, *Croton* oil, phenol and histamine.

#### 2.3.4. Gastroprotective Activity

The gastroprotective effect of *C. coriaceum* has been demonstrated by different authors. Leite et al. [45] found that, at a dose of 200 mg/Kg, the oil from the *C. coriaceum* fruit pulp was able to inhibit 60.5% of the gastric mucosal lesions induced by ethanol in Swiss mice (*M. musculus*). To achieve these objectives, the animals were deprived of their food and were induced to ingest, 12 h after this period, *C. coriaceum* oil at the concentration (200 and 400 mg/kg) or vehicle (tap water 10 mL/kg, control). One hour later, each animal was orally given 0.2 mL of ethanol (96%), and the animals were killed 30 min later. Their stomachs were excised and opened along the greater curvature, and the mucosal lesion area was measured by planimetry with a transparent grid placed on the surface of the glandular mucosa. Quirino et al. [46] also evidenced the pharmacological effect of *C. coriaceum* pulp oil on ethanol-induced ulcers and demonstrated that the activity involves α2-receptor mechanisms, endogenous prostaglandins, nitric oxide and K+ATP.

The gastroprotective effect is not restricted to the pulp oil but extends to the leaves of *C. coriaceum*, as demonstrated by Lacerda-Neto et al. [56]. According to these authors, the oral administration of the hydroethanolic extract of *C. coriaceum* leaves, at a dose of 100 mg/Kg, reduced gastric lesions by up to 86%. This study also demonstrated that opioid receptors, α2-adrenergic receptors and capsaicin-sensitive primary afferent neurons were involved in the gastric protection mechanism used by the extract of pequi leaves.

#### 2.3.5. Other Activities

Oliveira et al. [49] found that the oil of *C. coriaceum* at a dose of 400 mg/kg showed an antinociceptive effect in rats when compared to the control group. The effect found was similar to that of dexamethasone, a corticosteroid widely used to treat symptoms associated with various diseases. Experimentally, *C. coriaceum* oil was diluted in 2% Tween 80 concentrations and administered 45 min before induced arthritis or for 7 consecutive days by oral gavage at doses of 100, 200 and 400 mg/kg. Induced arthritis was performed using zymosan, which was administered as an injection 24 h after the last administration of the oil. In addition, there was a negative control, which received only saline and Tween 80 (2%). Another group was treated with the standard drug (dexamethasone) in each single-dose experimental trial 2 h before arthritis induction. The experimental groups were divided into subgroups to assess joint disability and joint swelling. Finally, the fluid and synovial tissue were collected for the evaluation of leukocyte recruitment, myeloperoxidase activity and cytokine release and immunohistochemistry. Changes in vascular permeability were assessed by the extravasation of Evans Blue dye into joint tissue 6 h after zymosan injection.

Another pharmacological effect for the oil from the fruits of *C. coriaceum* is its hypolipidemic action. Figueiredo et al. [63] treated Wistar rats (*Rattus norvegicus*) for 15 days with the fixed oil of “pequi” and subsequently induced dyslipidemia (elevation of cholesterol and triglycerides in plasma) through the administration of Triton WR-1339 (Tyloxapol). As a result, they showed that, at a dose of 2 g/kg, the oil was capable of reducing serum cholesterol levels by 16% and serum triglyceride levels by 23%. Another effect observed was a significant increase in the levels of HDL-C in the rats. The fixed oil of *C. coriaceum* was also able to prevent lung injury in rats subjected to short-term exposure [64].

Oliveira et al. [57] evaluated the effect of *C. coriaceum* oil in the treatment of seizures. In general, the authors first administered the fixed oil from the pulp of *C. coriaceum* to rats in increasing doses (25, 50 or 100 mg/kg). Subsequently, the animals were induced to seizure by PTZ and were immediately transferred to a cubic glass arena and video monitored for 15 min for the appearance of seizures. Unfortunately, no anticonvulsant action was found, but the administration at a dose of 100 mg/Kg was able to increase the latency to the first myoclonic spasm and the first generalized tonic-clonic seizures induced by pentylenetetrazol. These authors also demonstrated that the oil did not cause any significant adverse behavioral effect by means of the open field, rotarod, forced swimming and object recognition tests.

Regarding the antiparasitic effect, Alves et al. conducted the following experiment [54]. Antiparasitic activity was performed in 24-well microtiter plates, each well containing 1000 μL of 199 culture media supplemented with 1 × 10^6^ stationary phase promastigote forms with or without the extracts of interest at final concentrations of 0.1, 0, 05 and 0.025 mg/mL. After growth in contact with *C. coriaceum* extracts, viable cells were counted in a Neubauer chamber after 24, 48 and 72 h of treatment. They found that the ethanolic extracts of the bark and fruit pulp of *C. coriaceum* were effective against promastigote forms of *Leishmania* (*Leishmania*) *amazonensis* (MHOM/BR/1989/166MJO), with IC_50_ values of 38 and 30 µg/mL, respectively. These results were similar to those verified for the positive controls (pentamidine and meglumine antimoniate). These authors also concluded that the extracts showed anti-acetylcholinesterase activity compared to physiostigmine.

Tomiotto-Pellissier et al. [65] verified the leishmanicidal activity of *C. coriaceum* leaf extracts against *Leishmania* (L.) *amazonensis*. These authors used promastigote forms of *L. amazonensis* (106 cells/mL), which were subjected to different concentrations (25, 50 and 100 μg/mL) of extracts from the extracts of *C. coriaceum*. After the growth of 24, 48 and 72 h of treatment, the parasites were counted in a Neubauer chamber. From 24 h, all of the concentrations tested showed a significant reduction in the proliferation of *L. amazonensis* compared to the control (amphotericin B) or vehicle. The extracts also induced the loss of mitochondrial membrane potential, the production of reactive oxygen species (ROS), plasma membrane damage and phosphatidylserine exposure in promastigotes. Most parasites entered a process of late apoptosis.

Tomiotto-Pellissier et al. [66] first reported that the extracts of *C. coriaceum* fruit pulp and peel could induce the death of promastigote forms of *L. amazonensis* through apoptosis-like mechanisms and were also active against amastigote forms by activating the Nrf2/HO-1/ferritin pathway, reducing iron availability for parasite survival. In general, to identify such mechanisms, the authors treated the protozoa with extracts from the pulp and peel of the fruits of *C. coriaceum* 50 µg/mL during a period of 24 h, which were later processed and analyzed in scanning electron microscopy in order to assess whether the products caused any changes in the cells.

Despite the widespread popular use of plant products as therapeutic alternatives, there is evidence that they can be potentially toxic. In this sense, some authors have evaluated the in vitro and in vivo toxic effects of products obtained from *C. coriaceum*. Alves et al. [54], for example, evaluated in vitro the cytotoxic effect of the extracts of the bark and fruit pulp of *C. coriaceum* against macrophages. Experimentally, the viability of peritoneal macrophages treated with *C. coriaceum* extracts was evaluated using the 3-(4,5-dimethylthiazol-2-yl)-2,5-diphenyltetrazolium bromide (MTT) assay. BALB/c peritoneal macrophages (5 × 10^5^ U/mL) were cultured on media plates for 2 h for adherence at 37 °C and 5% CO_2_. The cells were washed with PBS, and then adherent cells were incubated with different concentrations of extracts (2.5–0.025 mg/mL) or with the vehicle (0.1% DMSO) and kept in the culture for 24 h under the previous conditions. After incubation with the extracts, the macrophages were washed with PBS, and MTT was added to a final concentration of 5 µg/mL in each well, followed by incubation for 4 h and the conditions mentioned above. The MTT formazan product was solubilized with 300 µL of DMSO, and the plates were read at 570 nm in a spectrophotometer. They verified moderate cytotoxicity, with IC_50_ values of 454 and 253 µg/mL, respectively. In contrast, against human erythrocytes, these products show low hemolytic activity at a concentration of 500 µg/mL.

In order to assess toxicity, Duavy et al. [52] tested the effect of the ethanolic and aqueous extract of the leaves of the species under study against the microcrustacean *Artemia salina* Leach. For that, the nauplii were exposed to different concentrations of the respective extracts (1–1000 µg/mL) during a period of 24 h. After this period, the surviving larvae were read. Concomitantly, a positive control group was prepared with aquamarine and potassium dichromate (K_2_Cr_2_O_7_), and a negative control was prepared with aquamarine and Tween 80. The ethanolic and aqueous extracts of *C. coriaceum* leaves presented, in vivo, a high toxicity on *A. salina* nauplii, with LC_50_ values of 14.9 and 18.5 µg/mL, respectively—more toxic than the positive control (potassium dichromate), whose LC_50_ was 55.9 µg/mL [52]. However, when using *Drosophila melanogaster* as a model, these authors found no toxicity of the aqueous extract at the concentration of 5 mg/mL during 5 days of the experiment [50]. In this trial, flies were fed a corn diet supplemented or not supplemented with *C. coriaceum* fixed oil in varying concentrations (1, 5 or 10 mg/g) for 7 days.

The allelopathic and larvicidal potential of *C. coriaceum* have also been reported [67,68]. Silva et al. [67] demonstrated that aqueous extracts of the leaves, fruits and stem of *C. coriaceum* possess allelochemicals capable of interfering with the germination of the seeds of *Lactuca sativa* L. (lettuce). These researchers evaluated the action of aqueous extracts at concentrations of 25, 50, 75 and 100% of the stem, leaf and fruit of *C. coriaceum* on the germination and growth of *L. sativa*. For that, they distributed the seeds of the donor species. The seeds were distributed in Petri dishes and spaced in order to facilitate the individual evaluation. Subsequently, they were placed in a B.O.D.-type chamber and were evaluated daily. Finally, 20 seedlings were randomly selected to obtain the averages referring to the length of the aerial and underground parts. Moreover, the extracts were able to interfere in the growth of the roots and stem of the recipient species. Regarding the larvicidal effect, Azevedo et al. [68] reported that *C. coriaceum* oil caused 100% mortality of *Aedes aegypti* larvae in a period of 120 h. In this study, methodologically, the researchers used the fixed oil from the fruits of *C. coriaceum*. They evaluated the larvicidal effect at different concentrations of the product (500, 1000, 1500, 2000 and 2500 ppm), using Tween^®^ 20 as a surfactant to aid in the dilution of oils in water. Ten *A. aegypti* larvae were used in instar L3 for each repetition. The larvae were submitted to treatments for 24, 48, 72, 96 and 120 h of exposure. Mortality was assessed when the dead larvae did not react to the mechanical stimulation of fine-tipped forceps.

### 2.4. Phytochemistry

The fruit and seeds of *C. coriaceum* are an important source of fixed oil (Table 2). Costa et al. [47], Figueiredo et al. [63] and Borges et al. [69] found that the pulp of the fruit of *C. coriaceum* is composed mostly of unsaturated fatty acids (>60%). Among the predominant unsaturated fatty acids, oleic acid (C18:1) stands out, while palmitic acid (C16:0) predominates among the saturated acids. In the lipid profile of the seeds, saturated fatty acids were found that were not present in the pulp, such as methyl-18-methylnonadecanoate (C20:0), docosanoic acid (C22:0) and lignoceric acid (C24:0) (Serra et al., 2020). On the other hand, arachidic acid (C20:0) and linolenic acid (C18:3), present in the fruits, were not detected in the seeds [70].

Besides compounds of the primary metabolism, *C. coriaceum* presents a wide variety of secondary compounds. Simple phenolics, flavones, flavonols, flavonones, flavononols, xanthones, tannins (pyrrobalic and hydrolysable), saponins, leucoanthocyanidins, catechins, steroids and alkaloids have been identified in *C. coriaceum* leaves [52,56,59,65,71].

The phytochemical studies of the phenolic compounds of *C. coriaceum* have concentrated on the fruits, bark and leaves (Table 2), with no research to date aimed at the bioprospecting of its roots and flowers. Among the compounds of a phenolic nature identified in the species, rutin, quercetin, epicatechin, isoquercitrin, gallic acid, chlorogenic acid, caffeic acid and ellagic acid stand out [50,54,55,59]. Such chemical heterogeneity may be responsible for the different activities verified for *C. coriaceum*.

Although numerous volatiles have been identified in C. brasiliensis [72,73,74], to our knowledge, there is no research targeting these constituents in *C. coriaceum*.

**Table 2 plants-11-01685-t002:** Substances identified in *Caryocar coriaceum*.

Compounds	Structure	Product	Plant Source
**Fatty acids**			
Palmitic acid [46,47,58,61,63,64,70,75,76,77,78,79]	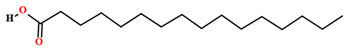	Fixed oil	Fruits (internal mesocarp and endocarp) and seeds
Oleic acid [46,47,58,61,63,70,75,76,77,78,79]	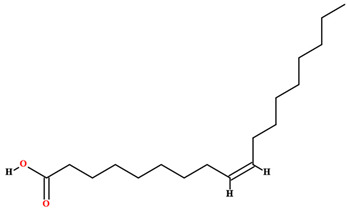	Fixed oil	Fruits (internal mesocarp and endocarp) and seeds
Stearic acid [39,40,54,56,64,70,75,76,77,78]	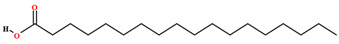	Fixed oil	Fruits (internal mesocarp and endocarp) and seeds
Palmitoleic acid [45,47,63,70,77,79]	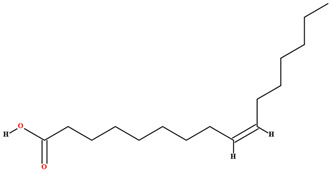	Fixed oil	Fruits (internal mesocarp and endocarp) and seeds
Linoleic acid [46,47,61,63,64,70,76,77,78,79]	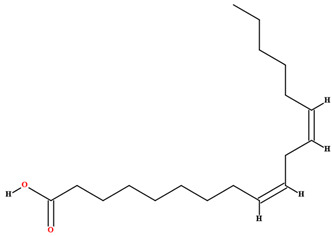	Fixed oil	Fruits (internal mesocarp and endocarp) and seeds
Heptadecenoic acid [46,47,61,63,79]	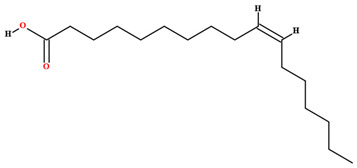	Fixed oil	Fruits (internal mesocarp and endocarp) and seeds
Eicosenoic acid [46,47,63,64,79]	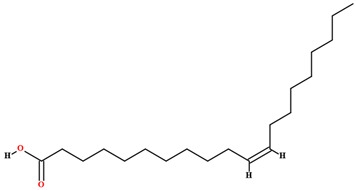	Fixed oil	Fruits (internal mesocarp and endocarp) and seeds
Myristic acid [70,76,78]	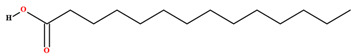	Fixed oil	Fruits (internal mesocarp); Seeds
Arachidic acid [70]	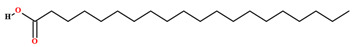	Fixed oil	Fruits (internal mesocarp)
Linolenic acid [58,70]	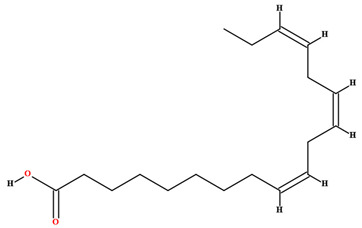	Fixed oil	Fruits (internal mesocarp)
Methyl-18-methylnonadecanoate [64]	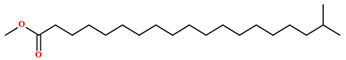	Fixed oil	Seeds
Docosanoic acid [64]	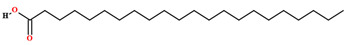	Fixed oil	Seeds
Lignoceric acid [64]	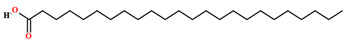	Fixed oil	Seeds
Heneicosanoic acid [58]	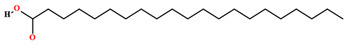	Fixed oil	Seeds
**Phenolic compounds**			
Quercetin [50,54,55,59]	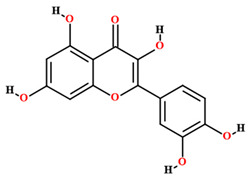	Ethanol, hydroethanolic, methanolic and aqueous extract	Fruits (internal mesocarp); leaves
Rutin [50,54,55,59]	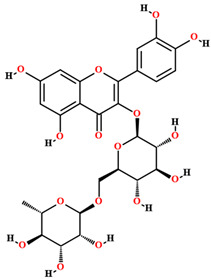	Ethanol, hydroethanolic, methanolic and aqueous extract	Fruits (internal mesocarp); bark; leaves
Catechin [50]	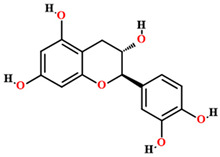	Aqueous extract	Leaves
Epicatechin [50]	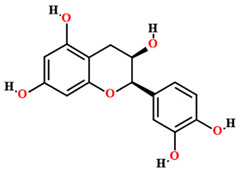	Aqueous extract	Leaves
Isoquercitrin [50]	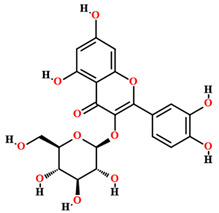	Aqueous extract	Leaves
Gallic acid [50,55,59]	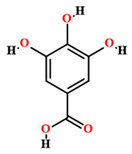	Ethanol, hydroethanolic, methanolic and aqueous extract	Leaves
Chlorogenic acid [50,55,59]	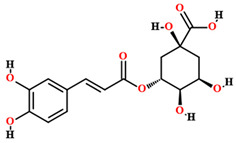	Ethanol, hydroethanolic, methanolic and aqueous extract	Leaves
Caffeic acid [50,55,59]	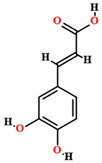	Ethanol, hydroethanolic, methanolic and aqueous extract	Leaves
Ellagic acid [50]	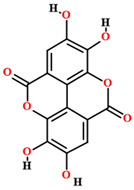	Aqueous extract	Leaves

### 2.5. Extractivism and Social, Economic and Cultural Importance

The fruits of *C. coriaceum* are of extreme socioeconomic importance for families of extractivist communities in the Chapada do Araripe, constituting up to 80% of the total family income. For this, the work is hard, since the extractivism of this fruit includes a set of activities such as collection, transport, processing and marketing either “in natura” or for its derivatives (oil), in which all family members participate [19,80,81,82,83,84,85].

For the extractive activity, initially, the families set up camps at the beginning of December around the forest to facilitate collection. Other camps are set up near the highways to sell the fruit to drivers passing through. The best known camps are located in the municipality of Jardim, receiving families from different districts of the Araripe region [19,86,87]. The camps are simple constructions of wood and clay and, in some cases, are constructed only with straw, with rare exceptions of brick. Most camps bring risks to the health of the inhabitants, such as the contact with insect vectors of *Trypanosoma cruzi*, the cause of Chagas disease. Due to the distance of the camps from the headquarters of the municipality, there is no electricity and drinking water in these places, and the only source of drinking water is the tanker trucks that pass by, supplying them every other fifteen days [19].

After settling in the camps in forest areas, the process of fruit gathering begins. This activity is carried out by all family members, with the women and children collecting in the areas closest to the camp and the men, due to their physical resistance, burrowing into the closed forests to harvest the fruit [19,81]. The peak of fruitfall occurs at 03:00 p.m., probably due to a higher production of ethylene gas that is strictly related to the high temperature in the environment (personal observation). Thus, the collection occurs the next day, around 4:00 a.m., due to a maximization of the fallen fruits and because, at this time, the climate is milder. It is worth mentioning that the fruit cannot be picked directly from the tree, since the product is still immature and of inferior quality to fruit picked from the ground. Thus, the community establishes certain rules for fruit collection, such as collecting only the fallen fruit and not shaking the trees [19,82,86]. However, there are extractivists who occasionally do not follow the rules, collecting immature fruits directly on the plant when they do not find them on the ground [82,86].

After collection, the pickers walk to the camps or transport the bags full of “pequi” on bicycles, motorbikes or even cars [16,19] (Figure 3). In the camps, the extractivists initiate the classification of the fruits according to their size, the small being designated as “choice” and the large ones designated as “chosen” The larger fruits are intended for immediate marketing, and the smaller ones for processing for oil production [82,88].

Processing the fruit for oil production is done by means of a technique called “rolling”. In this technique, a sharp-surfaced object, usually a knife, rotates around the fruit so that its skin is cut in two. Care must be taken not to cut the seeds, which are removed from the fruit with light pressure. This activity is performed by all family members, except children, due to the risk of accidents, but they can observe the procedure in order to learn the technique [81,82]. After “rolling”, the fruit peels can be used as fodder for animals (pigs and cattle) or as fertilizers. Occasionally, some extractivists discard the peels in the vicinity of the camps, causing the propagation of insects and other pests. The fruit “stones”, necessary for oil production, are stored in straw baskets until the moment of the oil extraction [81,89].

To obtain oil, forest wood resources are needed to act as fuel. The men are responsible for obtaining the wood and obtaining the oil, due to their physical resistance, and they enter the forest in search of dry wood. Some families extract the oil using liquefied gas, but due to its high price on the commercial market and the yield being significantly lower, this practice is not so common. It is worth noting that green wood, besides being forbidden to be obtained in the forest, is not very combustible [19,81,90].

The oil production process is arduous, as the first stage can take up to 5 h of cooking until a brownish mixture is obtained and the inner mesocarp becomes soft. After this process, the stones are rubbed with the help of a grater, which is made up of a wooden handle and a metal cylinder full of sharp points, in order to separate the mesocarp from the spiny epicarp (Figure 4). The work consists of repetitive movements inside the boilers, providing at the end of the process a darker brown oil with a pasty consistency [81].

After this process, the stones that are not pulped are removed from the boilers using a handmade skimmer. Before being discarded, they are washed with drinking water, and the solution returns to the boiler. This, in turn, continues the cooking process in order to agglutinate the oil. To do so, the extractors spend five hours continuously stirring the solution in the boiler with the help of a piece of wood, usually from the *C. coriaceum* trunk itself. This phase is very important for a better quality of the oil, because it can end up being cooked for a longer time than that which is ideal [81,90].

Finally, through agglutination, dark yellow spots appear, creating a superficial layer of oil in the boiler solution. With due care, the oily product is separated from the aqueous solution, brought back to the fire for two hours and subsequently filtered and bottled so that it can be marketed or used by the families themselves [81,90]. It is worth noting that the oil can be extracted from the stone (internal mesocarp + endocarp + almond) or from the kernel. The final difference is in the quality of the product, because the oil from the kernel, according to producers, is purer because it has a lighter color. However, this extraction is more arduous because the epicarp is of the thorny type, which can cause accidents during cutting and handling. Due to this set of factors, the oil from the almonds has a higher cost in the commercial market [81].

The marketing of oil is driven mainly by the pilgrimages, which are pilgrimages to religious sites or places of devotion and occur annually in the city of Juazeiro do Norte in devotion to Father Cícero (1844–1934), a great religious and political leader of the region [90,91,92]. Among the pilgrimages that are more profitable are those that occur in the off-season period, such as the Anniversary of the Death of Father Cícero (20 July), the Pilgrimage of Our Lady of Sorrows (15 September), the Pilgrimage of Saint Francis of the Wounds (4 October) and the Pilgrimage of The Day of The Dead (30 October to 1 November). These pilgrimages attract thousands of faithful from all over Brazil, who buy *C. coriaceum* products for medicinal, food and commercial purposes [87,90,93,94,95,96].

At the end of every *C. coriaceum* harvest, important cultural events take place in the region, such as the “Pequis Festival”. This festival takes place over two days, usually in March. During this period, there is a Holy Mass of Thanksgiving, cattle-catching competitions and regional musical attractions (“forró”). Such events are used to commemorate and give thanks for another year of harvesting, as well as to celebrate and bring together friends and family [30,81,90]. Traditionally, recipes are prepared using the fruits of *C. coriaceum*, such as baião de dois (a combination of rice and beans), munguzá (a food that uses corn, beans, pork and vegetables) and the famous pequizada, which is prepared with milk, spices and *C. coriaceum* [30,90].

### 2.6. Conservation of Caryocar Coriaceum

According to the International Union for Conservation of Nature and Natural Resources (IUCN) Red List, *C. coriaceum* is an endangered plant species [31]. Such conservation status is due to a number of factors, such as increasing extractivism, slow germination, the reduction of dispersing animals, deforestation and fires in the Cerrado [10,24,88].

The fruits of *C. coriaceum* suffer great anthropic pressure because they are highly appreciated in cuisine and popular medicine. To meet the demand, thousands of fruits are used to produce oil and to be sold in natura. Thus, if there is no proper control of extractivism in accordance with the replacement standards, this overexploitation may lead to the collapse of these natural resources, because, with the extractive pressure and the reduced amount of diaspores at the end of the harvest, there is a low recruitment of seedlings, which already present a low frequency and restricted distribution in the ecosystem [24,97]. In addition to these factors, the seeds of *C. coriaceum* present a slow germination. Such phenomenon is due to the endogenous dormancy that the seeds present, which can take a year to germinate. This slow and delayed germination further aggravates the conservation of the species [10,85].

The reduction of dispersing animals of *C. coriaceum* diaspores is a rarely reported factor, but it is one that deserves attention. The process of the fruit dispersal of this species is an essential step in the regeneration of individuals, as well as in the biological maintenance of natural environments [98]. Among the natural dispersers are the beetles of the Scarabaeidae family, which are known as “dung beetles”. These insects use the decomposing pulp of *C. coriaceum* fruits as a food resource, thus reducing the thorny endocarp, besides burying the fruits, favoring seed germination [88,98,99]. Unfortunately, these arthropods, on account of having a sedentary life, are more vulnerable to climate change, and their local distribution is strongly influenced by vegetation cover. As the Chapada do Araripe has been constantly targeted by anthropic actions such as deforestation and burning, the populations of these beetles tend to decrease [99,100,101].

Besides insects, there are vertebrates that participate in the dispersal of *C. coriaceum*, such as *Dasyprocta prymnolopha* Wagler, 1831 (Dasyproctidae), popularly called “cutia”. This rodent helps in the dissemination of seeds, bringing benefits such as reducing attacks by natural predators, assisting in the colonization of new environments and increasing genetic variability over space [98]. Unfortunately, these mammals are targets of hunters in the region, which are used for food purposes [102]. With the above, it is evident that forest fragmentation, associated with hunting, has reduced the local fauna and, consequently, the dispersers of *C. coriaceum* fruits [98,103].

Finally, deforestation and burning are the main factors that accelerate this process of extinction of the species. Such criminal practices are used with the intent of obtaining pasture areas for livestock. One of the largest fires in Chapada do Araripe occurred in early 2020, which affected about two thousand hectares of forest. This destruction had a direct impact on the production of the fruit, with 80% of the production being affected [83,104].

It is clear that *C. coriaceum* receives great anthropic pressure, and measures are necessary to ensure that the extinction of the species does not occur. As a strategy, there are numerous measures that can be taken—for example, reducing the exploitation of the fruits. The excessive collection of fruits prevents the propagation of new individuals and the regeneration of populations in deforested areas. The creation of public policies that focus on the protection of *C. coriaceum*-dispersing animals, as well as on greater forest surveillance and the planting of seedlings of the species in several areas of the fragmented forest, should also be considered [87,88].

One of the advances in conservation was the establishment of law 9.985 in 2000 by the Brazilian Institute of the Environment and Renewable Natural Resources (IBAMA), which prohibited practices or any activities that impede the natural regeneration of ecosystems. Among the prohibited activities is the entry of cattle into the forest, an activity widely used by extractivists, because the animals, by opening paths in the forest, facilitate the collection of the fruits of *C. coriaceum* [81].

## 3. Materials and Methods

### 3.1. Search in Databases and Inclusion and Exclusion Criteria

The keyword “*Caryocar coriaceum*” was used as the main index for searching the following platforms: PubMed^®^, PubMed Central^®^, SciElo, Scopus^®^ and Web of Science^TM^. Articles published in English and Portuguese in the last 39 years (1983–2022) were considered. Abstracts published in conference proceedings, course completion papers, dissertations and theses and those without mention of the ethnomedicinal uses, biological activities, phytochemistry, extraction and conservation of *C. coriaceum* were discarded.

### 3.2. Data Screening and Categorization of Information

A total of 215 scientific documents were found in the databases searched. After an initial analysis, 75 documents were excluded for duplicity. A total of 140 documents were analyzed in full (title, abstract, keywords and full text), 33 being excluded for not meeting the scope of the review. A total of 107 documents were used in the qualitative analysis, and another 12 were used to compose the present review (Figure 5). Information management was performed with the help of Mendeley Reference Manager^©^ software version 2.67.0 (Mendeley Ltd., London, UK).

The information found was categorized into: (1) botanical aspects and geographical distribution; (2) ethnobotanical uses; (3) biological and pharmacological activities; (4) phytochemistry; (5) extraction and social, economic and cultural importance; and (6) conservation of *Caryocar coriaceum*.

## 4. Conclusions and Future Prospects

*Caryocar coriaceum* is a tree of great economic, cultural and medicinal importance for the inhabitants of the Chapada do Araripe and neighboring regions. Such importance is noted for the use of the fruits and its by-products such as oil. In this review, we demonstrate that the fixed oil of *C. coriaceum* is widely used in folk medicine in communities in the Chapada do Araripe. Such use is supported by pharmacological tests, which show that the fatty acids present in the natural product are possibly responsible for the pharmacological property, one of the reasons for having patents with such a product. However, even though it is a very relevant species and is known for its therapeutic effects, it is still rarely studied by the scientific community when compared to another species of the same genus, *C. brasiliense*—also known as “pequi”.

## Figures and Tables

**Figure 1 plants-11-01685-f001:**
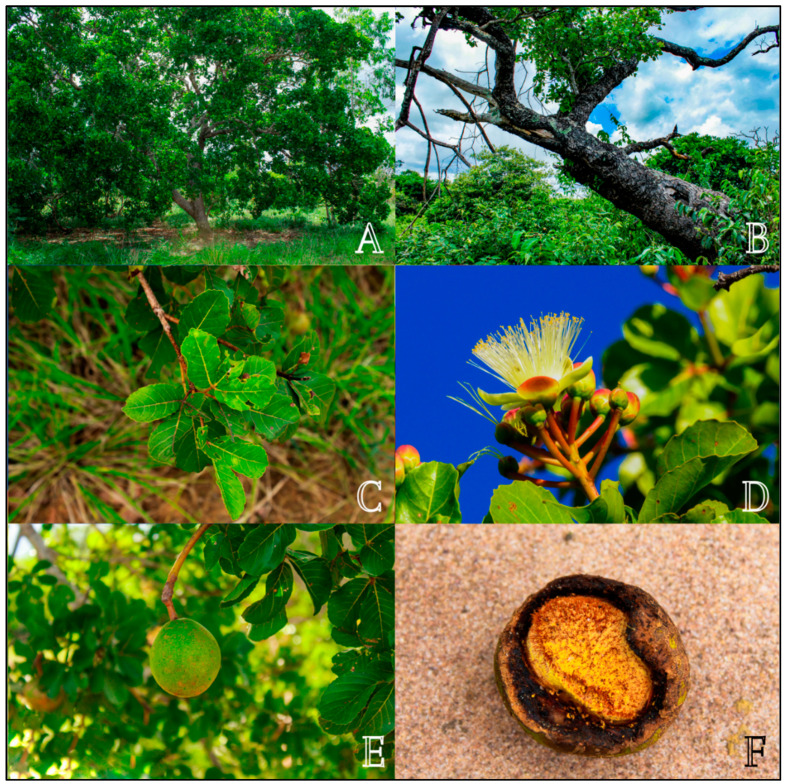
General aspect of Caryocar coriaceum Wittm. (Caryocaraceae). (**A**) = Habit; (**B**) = Stem; (**C**) = Leaves; (**D**) = Flower and floral buds; (**E**) = Fruit; (**F**) = Fruit with endocarp exposed after the removal of part of the epicarp by ants.

**Figure 2 plants-11-01685-f002:**
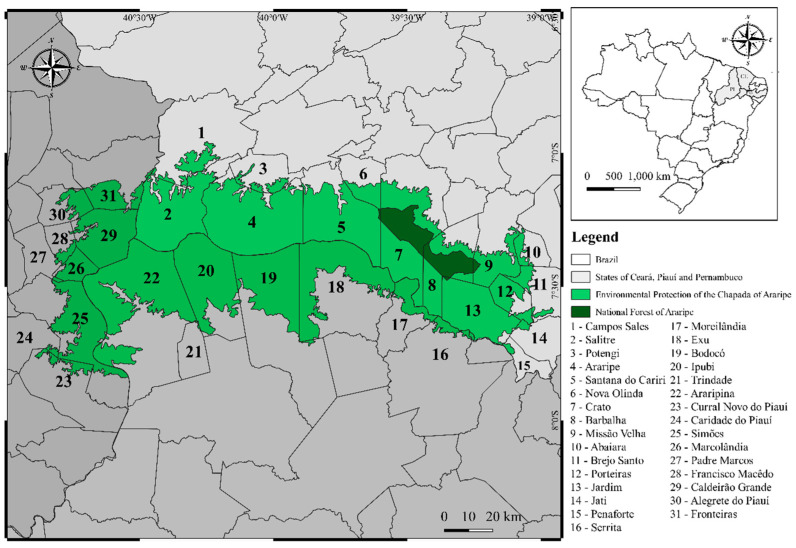
Map of the Araripe Environmental Protection Area (APA—Araripe, Ceará, Brazil) where *Caryocar coriaceum* Wittm. (Caryocaraceae) occurs spontaneously.

**Figure 3 plants-11-01685-f003:**
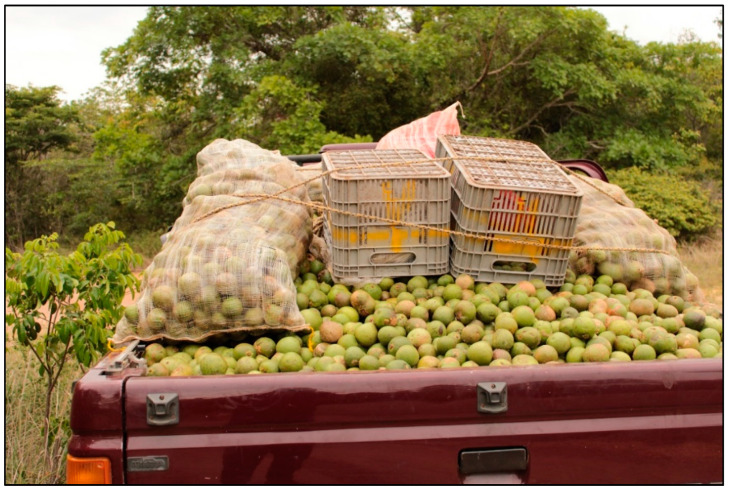
Car commonly used to transport *Caryocar coriaceum* Wittm. during the harvest.

**Figure 4 plants-11-01685-f004:**
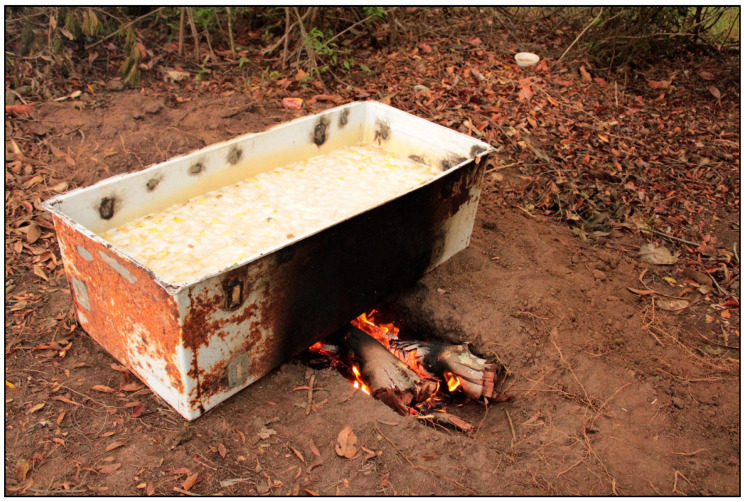
Process of cooking the fruits of *Caryocar coriaceum* Wittm. in boilers for the production of oil at the Barreiro Novo camp, Jardim—CE, Brazil.

**Figure 5 plants-11-01685-f005:**
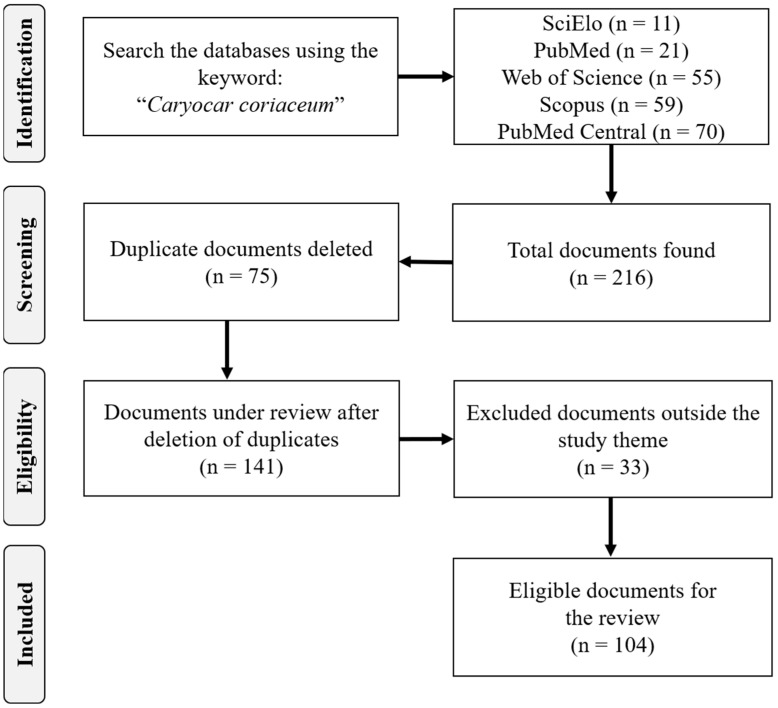
Flowchart describing the search and selection strategies.

## Data Availability

No new data were created or analyzed in this study. Data sharing is not applicable to this article.

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
