# Peer review of "Caryocar coriaceum Wittm. (Caryocaraceae): Botany, Ethnomedicinal Uses, Biological Activities, Phytochemistry, Extractivism and Conservation Needs"

_plants, 2022, doi:10.3390/plants11131685_

Round 1

Reviewer 1 Report

Comments

Although the article is informative, the titles and subtitles are haphazardly given. So it should be rewritten perfectly. For example, write

Introduction (characters, distribution, origin, history etc)

Methods (searched words, datasource etc)

Results and discussion: subheading like Ethnomedicinal, Chemicals composition and Biological activities (with sub headings). Toxicity if present.

Conclusions

Some specific comments:

Abstract: Rewrite the abstract. Please write the searched words and also write the datasourcein very short form.

Introduction: In first and second paragraph, do not write about history and other families. Write specifically about your research species. Also include more literatures.

In section 2.2. Ethnopharmacological uses: Write specifically about species not other things.

In section the antimicrobial, healing, anti-inflammatory, gastroprotective antioxidant potential activities, although less informative, but should be written in clear cut details of testing animal, dozes and results.

The section other activities should be further sub headed and detailed should be written.

Section: Extractivism and social, economic and cultural importance is unnecessarily long. Rewrite and concise it.

Rewrite and concise Conservation of Caryocar coriaceum section.

I donot understand why some parts are highlighted.

Author Response

Although the article is informative, the titles and subtitles are haphazardly given. So it should be rewritten perfectly. For example, write Introduction (characters, distribution, origin, history etc), Methods (searched words, datasource etc), Results and discussion: subheading like Ethnomedicinal, Chemicals composition and Biological activities (with sub headings). Toxicity if present. Conclusions

Dear reviewer, we were unable to understand your request. We don't know which file arrived for your review. But no topic is added at random. If you look at the file, the subtitles you are requesting are already added.

Some specific comments:

Abstract: Rewrite the abstract. Please write the searched words and also write the datasourcein very short form.

Dear reviewer, we added to the abstract the word used in the review search, as well as the searched databases. As for rewriting the abstract, we would like to know specifically what to rewrite as it addresses the requirements of Plants journal.

Introduction: In first and second paragraph, do not write about history and other families. Write specifically about your research species. Also include more literatures.

Dear reviewer, as requested, we removed the paragraphs that are not directly associated with the species. About including literatures, it's subjective and we can't understand. Could you tell us the list of this literature? Because our focus is on Caryocar coriaceum, when we include other literatures, we will be avoiding your request when writing about the research species.

In section 2.2. Ethnopharmacological uses: Write specifically about species not other things.

Accepted request. However, for scientific writing purposes, we leave in the first paragraph of the introduction, some information that we think is relevant.

The section other activities should be further sub headed and detailed should be written. Section: Extractivism and social, economic and cultural importance is unnecessarily long. Rewrite and concise it.

This section has been reduced in order to meet the requests, however, we keep the most important parts, which, if removed, would leave the manuscript incomplete and confusing.

Rewrite and concise Conservation of Caryocar coriaceum section.

Dear reviewer, we would like to ask permission not to modify this section in a shortened way. Unfortunately, Brazil is currently undergoing a policy against the environment, so we are rapidly losing large areas of forests. One of our struggles, as conservationists, is to expose such issues to the entire scientific community. We would love for you to understand us at this point.

I do not understand why some parts are highlighted.

The article returned for revisions, and we were asked to highlight the changes made in yellow.

Reviewer 2 Report

This is an interesting review on Caryocar coriaceum. The authors reviewed studies conducted with the species highlighting its ethnomedicinal use, its pharmacological potential, its chemical constituents, and its cultural and socioeconomic importance. However, the review was not well presented, and considerable revisions were suggested as below.

1. The abstract was found to be insufficiently informative, and insufficiently attractive; the most important findings and the prospective should be added, and some very common information should be deleted. The whole manuscript should also undergo major scientific revision to systematically describe and summarize the major findings in the past.

2. The introduction should be more specific and more informative to summarize the-state-of-the-art research on Caryocar coriaceum, which should be significantly enhanced.

3. There were too few figures with poor scientific meaning. At least, two more figures can be added, one is used to depict the phytochemicals, ethno-medicinal uses, biological activities of different part of this valuable plant; the other is used to show the major chemical structures of major phytochemicals as shown in table 2. In addition, table 2 should be revised and omit the chemical structures as they are revised as figures.

4. In many places throughout the text, the use of references are problematic, such as blow, the authors put too many reference after a single sentence [17–25], please revise them and check all other similar ones in the context to make the citation more specific and meaningful.

“The trunks have thick bark, thick and angular branches, 81 which can grow to the sides of the plant or close to the ground, which set the species 82 apart from others in Cerrado areas [17–25].”

5. Based on table 2, I have some doubt on the phytochemical research and activity tests conducted on this plant, since most of the phytochemicals summarized in this table are known and very common natural products, I highly doubt the derived activity only based on these compounds, please have a double check on the related references.

6. Conclusions and future prospects should be significantly enhanced, and identify the major issues and propose the future efforts.

Author Response

This is an interesting review on Caryocar coriaceum. The authors reviewed studies conducted with the species highlighting its ethnomedicinal use, its pharmacological potential, its chemical constituents, and its cultural and socioeconomic importance. However, the review was not well presented, and considerable revisions were suggested as below.

  1. The abstract was found to be insufficiently informative, and insufficiently attractive; the most important findings and the prospective should be added, and some very common information should be deleted. The whole manuscript should also undergo major scientific revision to systematically describe and summarize the major findings in the past.

Thanks for the review. We added information from the data collection to the summary.

  1. The introduction should be more specific and more informative to summarize the-state-of-the-art research on Caryocar coriaceum, which should be significantly enhanced.

Dear reviewer, we have added a paragraph in the introduction to address your request.

  1. There were too few figures with poor scientific meaning. At least, two more figures can be added, one is used to depict the phytochemicals, ethno-medicinal uses, biological activities of different part of this valuable plant; the other is used to show the major chemical structures of major phytochemicals as shown in table 2. In addition, table 2 should be revised and omit the chemical structures as they are revised as figures.

Dear reviewer, we fully understand your request. We previously tried to make a figure with all the phytochemicals, however the image had many, it was low resolution. In this way, we ask for understanding so that the constituents stay within the table, in addition to being more didactic for the reader.

  1. In many places throughout the text, the use of references are problematic, such as blow, the authors put too many reference after a single sentence [17–25], please revise them and check all other similar ones in the context to make the citation more specific and meaningful. “The trunks have thick bark, thick and angular branches, 81 which can grow to the sides of the plant or close to the ground, which set the species 82 apart from others in Cerrado areas [17–25].”

Dear reviewer, unfortunately this is a reality in which we will not be able to interfere much. What happens is that several works report the same information, mainly in the botanical description section. We try our best to distribute references ethically, so as not to fail to cite the appropriate references.

  1. Based on table 2, I have some doubt on the phytochemical research and activity tests conducted on this plant, since most of the phytochemicals summarized in this table are known and very common natural products, I highly doubt the derived activity only based on these compounds, please have a double check on the related references.

Dear reviewer, this was a suggestion from a previous reviewer, which we also don't agree with, however he said it was a mandatory correction. As your consideration is similar to reviewer 3, we decided to remove such information from the table.

  1. Conclusions and future prospects should be significantly enhanced, and identify the major issues and propose the future efforts.

After the review, we added that request and included it in the conclusion.

Reviewer 3 Report

Section Ethnopharmacological uses, you give only ethnobotanical information, where the pharmacology in this section? Should be Ethnomedical uses?

Table 1 must only mention the biological activities of the isolated compounds from the plant, not in general, this can cause confusion. As an example, you mention Chlorogenic acid as hypoglycemic, but this propriety is not described for the plant.

The manuscript is a little bit long, I recommend shorten it, as an example the taxonomical information can be found in other published sources, maybe a short section.

Where are your own conclusions that correlate the ethno, the pharmacology and phytochemistry of the plant?

In general is a nice work

Author Response

Section Ethnopharmacological uses, you give only ethnobotanical information, where the pharmacology in this section? Should be Ethnomedical uses?

Dear reviewer, we have corrected the term for ethnomedical uses. Thanks for this correction.

Table 1 must only mention the biological activities of the isolated compounds from the plant, not in general, this can cause confusion. As an example, you mention Chlorogenic acid as hypoglycemic, but this propriety is not described for the plant.

Dear reviewer, this was a suggestion from a previous reviewer, which we also don't agree with, however he said it was a mandatory correction. In this way, we will be removing such information from the table.

The manuscript is a little bit long, I recommend shorten it, as an example the taxonomical information can be found in other published sources, maybe a short section.

Dear reviewer, we agree with your comment. This article is a little long, and we have shortened it in some sections, such as extractivism and conservation, in order to fulfill your wish. We would like to ask permission not to shorten the botanical description section, as this article may be the basis for readings that want a compilation of descriptions. We hope your lordship understands us.

Where are your own conclusions that correlate the ethno, the pharmacology and phytochemistry of the plant?

After the review, we added that request and included it in the conclusion.

Round 2

Reviewer 1 Report

Although my all-other review comments are well addressed, but

in section, 2.3. Biological and pharmacological activities and subheadings, the detailed methods are not written perfectly but the details of results were well written. See the reference below rewrite this section.

So please see the paper as a reference

Rokaya M.B., Parajuli B., Bhatta K.P., Timsina B. (2020) Neopicrorhiza scrophulariiflora (Pennell) Hong: A comprehensive review of its traditional uses, phytochemistry, pharmacology and safety. Journal of Ethnopharmacology 247, DOI: 10.1016/j.jep.2019.112250

Author Response

Although my all-other review comments are well addressed, but in section, 2.3. Biological and pharmacological activities and subheadings, the detailed methods are not written perfectly but the details of results were well written. See the reference below rewrite this section.

So please see the paper as a reference

Rokaya M.B., Parajuli B., Bhatta K.P., Timsina B. (2020) Neopicrorhiza scrophulariiflora (Pennell) Hong: A comprehensive review of its traditional uses, phytochemistry, pharmacology and safety. Journal of Ethnopharmacology 247, DOI: 10.1016/j.jep.2019.112250

Dear reviewer, we greatly appreciate your contributions to the enrichment of the review manuscript. As requested, we have added the methods used in section 2.3, mainly the more specific methods. However, it was not possible to add the methods of the more common methods because other reviewers were complaining about the size of the manuscript. We hope you can understand us, as we want to accept all possible suggestions.

Reviewer 2 Report

The authos have tried their best to address most of the comments raised by reviewers. The work is  improved, and acceptance is thus suggested.

Author Response

Dear reviewer,

Thanks for your works and overall constructive comments that markedly improved our manuscript.

Round 3

Reviewer 1 Report

Hence you have addressed my review comments, but still in section 2.3.5. Other activities: the details of experiments are lacking. It could be better if you write all.

Author Response

Dear reviewer, we appreciate your contributions. We have carefully added the requested methods in that section. All the best for you.